# Dropout with Expectation-linear Regularization

**Xuezhe Ma, Yingkai Gao**
Language Technologies Institute
Carnegie Mellon University
{xuezhem, yingkaig}@cs.cmu.edu

**Zhiting Hu, Yaoliang Yu**
Machine Learning Department
Carnegie Mellon University
{zhitinghu, yaoliang}@cs.cmu.edu

**Yuntian Deng**
School of Engineering and Applied Sciences
Harvard University
dengyuntian@gmail.com

**Eduard Hovy**
Language Technologies Institute
Carnegie Mellon University
hovy@cmu.edu

## Abstract

Dropout, a simple and effective way to train deep neural networks, has led to a number of impressive empirical successes and spawned many recent theoretical investigations. However, the gap between dropout's training and inference phases, introduced due to tractability considerations, has largely remained under-appreciated. In this work, we first formulate dropout as a tractable approximation of some latent variable model, leading to a clean view of parameter sharing and enabling further theoretical analysis. Then, we introduce (approximate) expectation-linear dropout neural networks, whose inference gap we are able to formally characterize. Algorithmically, we show that our proposed measure of the inference gap can be used to regularize the standard dropout training objective, resulting in an *explicit* control of the gap. Our method is as simple and efficient as standard dropout. We further prove the upper bounds on the loss in accuracy due to expectation-linearization, describe classes of input distributions that expectation-linearize easily. Experiments on three image classification benchmark datasets demonstrate that reducing the inference gap can indeed improve the performance consistently.

## 1 Introduction

Deep neural networks (DNNs, e.g., LeCun et al., 2015; Schmidhuber, 2015), if trained properly, have been demonstrated to significantly improve the benchmark performances in a wide range of application domains. As neural networks go deeper and deeper, naturally, its model complexity also increases quickly, hence the pressing need to *reduce overfitting* in training DNNs. A number of techniques have emerged over the years to address this challenge, among which dropout (Hinton et al., 2012; Srivastava, 2013) has stood out for its simplicity and effectiveness. In a nutshell, dropout *randomly* "drops" neural units during training as a means to prevent feature co-adaptation—a sign of overfitting (Hinton et al., 2012). Simple as it appears to be, dropout has led to several record-breaking performances (Hinton et al., 2012; Ma & Hovy, 2016), and thus spawned a lot of recent interests in analyzing and justifying dropout from the theoretical perspective, and also in further improving dropout from the algorithmic and practical perspective.

In their pioneering work, Hinton et al. (2012) and Srivastava et al. (2014) interpreted dropout as an extreme form of model combination (aka. model ensemble) with extensive parameter/weight sharing, and they proposed to learn the combination through minimizing an appropriate expected loss. Interestingly, they also pointed out that for a single logistic neural unit, the output of dropout is in fact the geometric mean of the outputs of the model ensemble with shared parameters. Subsequently, many theoretical justifications of dropout have been explored, and we can only mention a few here due to space limits. Building on the weight sharing perspective, Baldi & Sadowski (2013; 2014) analyzed the ensemble averaging property of dropout in deep non-linear logistic networks, and supported the view that dropout is equivalent to applying stochastic gradient descent on some regularized

loss function. Wager et al. (2013) treated dropout as an adaptive regularizer for generalized linear models (GLMs). Helmbold & Long (2016) discussed the differences between dropout and traditional weight decay regularization. In terms of statistical learning theory, Gao & Zhou (2014) studied the Rademacher complexity of different types of dropout, showing that dropout is able to reduce the Rademacher complexity polynomially for shallow neural networks (with one or no hidden layers) and exponentially for deep neural networks. This latter work (Gao & Zhou, 2014) formally demonstrated that dropout, due to its regularizing effect, contributes to reducing the inherent model complexity, in particular the variance component in the generalization error.

Seen as a model combination technique, it is intuitive that dropout contributes to reducing the variance of the model performance. Surprisingly, dropout has also been shown to play some role in reducing the model bias. For instance, Jain et al. (2015) studied the ability of dropout training to escape local minima, hence leading to reduced model bias. Other studies (Chen et al., 2014; Helmbold & Long, 2014; Wager et al., 2014) focus on the effect of the dropout noise on models with shallow architectures. We noted in passing that there are also some work (Kingma et al., 2015; Gal & Ghahramani, 2015; 2016) trying to understand dropout from the Bayesian perspective.

In this work, we first formulate dropout as a tractable approximation of a latent variable model, and give a clean view of weight sharing (§3). Then, we focus on an *inference gap* in dropout that has somehow gotten under-appreciated: In the inference phase, for computational tractability considerations, the model ensemble generated by dropout is approximated by a *single* model with scaled weights, resulting in a gap between training and inference, and rendering the many previous theoretical findings inapplicable. In general, this inference gap can be very large and no attempt (to our best knowledge) has been made to control it. We make three contributions in bridging this gap: Theoretically, we introduce expectation-linear dropout neural networks, through which we are able to explicitly quantify the inference gap (§4). In particular, our theoretical results explain why the max-norm constraint on the network weights, a standard practice in training DNNs, can lead to a small inference gap hence potentially improve performance. Algorithmically, we propose to add a sampled version of the inference gap to regularize the standard dropout training objective (*expectation-linearization*), hence allowing explicit control of the inference gap, and analyze the interaction between expectation-linearization and the model accuracy (§5). Experimentally, through three benchmark datasets we show that our regularized dropout is not only as simple and efficient as standard dropout but also consistently leads to improved performance (§6).

## 2    DROPOUT NEURAL NETWORKS

In this section we set up the notations, review the dropout neural network model, and discuss the inference gap in standard dropout training that we will attempt to study in the rest of the paper.

### 2.1    DNNS AND NOTATIONS

Throughout we use uppercase letters for random variables (and occasionally for matrices as well), and lowercase letters for realizations of the corresponding random variables. Let $X \in \mathcal{X}$ be the input of the neural network, $Y \in \mathcal{Y}$ be the desired output, and $D = \{(x_1, y_1), \ldots, (x_N, y_N)\}$ be our training sample, where $x_i, i = 1, \ldots, N$, (resp. $y_i$) are usually i.i.d. samples of $X$ (resp. $Y$).

Let $\mathbf{M}$ denote a deep neural network with $L$ hidden layers, indexed by $l \in \{1, \ldots, L\}$. Let $\mathbf{h}^{(l)}$ denote the output vector from layer $l$. As usual, $\mathbf{h}^{(0)} = x$ is the input, and $\mathbf{h}^{(L)}$ is the output of the neural network. Denote $\theta = \{\theta_l : l = 1, \ldots, L\}$ as the set of parameters in the network $\mathbf{M}$, where $\theta_l$ assembles the parameters in layer $l$. With dropout, we need to introduce a set of dropout random variables $S = \{\Gamma^{(l)} : l = 1, \ldots, L\}$, where $\Gamma^{(l)}$ is the dropout random variable for layer $l$. Then the deep neural network $\mathbf{M}$ can be described as:

$$\mathbf{h}^{(l)} = f_l(\mathbf{h}^{(l-1)} \odot \gamma^{(l)}; \theta_l), \quad l = 1, \ldots, L, \tag{1}$$

where $\odot$ is the element-wise product and $f_l$ is the transformation function of layer $l$. For example, if layer $l$ is a fully connected layer with weight matrix $W$, bias vector $b$, and sigmoid activation function $\sigma(x) = \frac{1}{1+\exp(-x)}$, then $f_l(x) = \sigma(Wx + b)$). We will also use $\mathbf{h}^{(l)}(x, s; \theta)$ to denote the output of layer $l$ with input $x$ and dropout value $s$, under parameter $\theta$.

In the simplest form of dropout, which is also called standard dropout, $\Gamma^{(l)}$ is a vector of independent Bernoulli random variables, each of which has probability $p_l$ of being 1 and $1 - p_l$ of being 0. This corresponds to dropping each of the weights independently with probability $p_l$.

## 2.2 DROPOUT TRAINING

The standard dropout neural networks can be trained using stochastic gradient decent (SGD), with a sub-network sampled by dropping neural units for each training instance in a mini-batch. Forward and backward pass for that training instance are done only on the sampled sub-network. Intuitively, dropout aims at, simultaneously and jointly, training an ensemble of exponentially many neural networks (one for each configuration of dropped units) while sharing the same weights/parameters.

The goal of the stochastic training procedure of dropout can be understood as minimizing an expected loss function, after marginalizing out the dropout variables (Srivastava, 2013; Wang & Manning, 2013). In the context of maximal likelihood estimation, dropout training can be formulated as:

$$\theta^* = \underset{\theta}{\operatorname{argmin}} \, \mathrm{E}_{S_D}[-l(D, S_D; \theta)] = \underset{\theta}{\operatorname{argmin}} \, \mathrm{E}_{S_D}\Big[ -\sum_{i=1}^{N} \log p(y_i|x_i, S_i; \theta)\Big], \qquad (2)$$

where recall that $D$ is the training sample, $S_D = \{S_1, \ldots, S_N\}$ is the dropout variable (one for each training instance), and $l(D, S_D; \theta)$ is the (conditional) log-likelihood function defined by the conditional distribution $p(y|x, s; \theta)$ of output $y$ given input $x$, under parameter $\theta$ and dropout variable $s$. Throughout we use the notation $\mathrm{E}_Z$ to denote the conditional expectation where all random variables except $Z$ are conditioned on.

Dropout has also been shown to work well with regularization, such as L2 weight decay (Tikhonov, 1943), Lasso (Tibshirani, 1996), KL-sparsity(Bradley & Bagnell, 2008; Hinton, 2010), and max-norm regularization (Srebro et al., 2004), among which the max-norm regularization — that constrains the norm of the incoming weight matrix to be bounded by some constant — was found to be especially useful for dropout (Srivastava, 2013; Srivastava et al., 2014).

## 2.3 DROPOUT INFERENCE AND GAP

As mentioned before, dropout is effectively training an ensemble of neural networks with weight sharing. Consequently, at test time, the output of each network in the ensemble should be averaged to deliver the final prediction. This averaging over exponentially many sub-networks is, however, intractable, and standard dropout typically implements an approximation by introducing a *deterministic* scaling factor for each layer to replace the *random* dropout variable:

$$\mathrm{E}_S[\mathbf{H}^{(L)}(x, S; \theta)] \overset{?}{\approx} \mathbf{h}^{(L)}(x, \mathrm{E}[S]; \theta), \qquad (3)$$

where the right-hand side is the output of a single deterministic neural network whose weights are scaled to match the *expected* number of active hidden units on the left-hand side. Importantly, the right-hand side can be easily computed since it only involves a single deterministic network.

Bulò et al. (2016) combined dropout with knowledge distillation methods (Hinton et al., 2015) to better approximate the averaging processing of the left-hand side. However, the quality of the approximation in (3) is largely unknown, and to our best knowledge, no attempt has been made to *explicitly* control this inference gap. The main goal of this work is to explicitly quantify, algorithmically control, and experimentally demonstrate the inference gap in (3), in the hope of improving the generalization performance of DNNs eventually. To this end, in the next section we first present a latent variable model interpretation of dropout, which will greatly facilitate our later theoretical analysis.

## 3 DROPOUT AS LATENT VARIABLE MODELS

With the end goal of studying the inference gap in (3) in mind, in this section, we first formulate dropout neural networks as a latent variable model (LVM) in § 3.1. Then, we point out the relation between the training procedure of LVM and that of standard dropout in § 3.2. The advantage of formulating dropout as a LVM is that we need only deal with a single model (with latent structure), instead of an ensemble of exponentially many different models (with weight sharing). This much

simplified view of dropout enables us to understand and analyze the model parameter $\theta$ in a much more straightforward and intuitive way.

## 3.1 AN LVM FORMULATION OF DROPOUT

A latent variable model consists of two types of variables: the observed variables that represent the empirical (observed) data and the latent variables that characterize the hidden (unobserved) structure. To formulate dropout as a latent variable model, the input $x$ and output $y$ are regarded as observed variables, while the dropout variable $s$, representing the sub-network structure, is hidden. Then, upon fixing the input space $\mathcal{X}$, the output space $\mathcal{Y}$, and the latent space $\mathcal{S}$ for dropout variables, the conditional probability of $y$ given $x$ under parameter $\theta$ can be written as

$$p(y|x;\theta) = \int_{\mathcal{S}} p(y|x,s;\theta)p(s)d\mu(s), \tag{4}$$

where $p(y|x,s;\theta)$ is the conditional distribution modeled by the neutral network with configuration $s$ (same as in Eq. (2)), $p(s)$ is the distribution of dropout variable $S$ (e.g. Bernoulli), here assumed to be independent of the input $x$, and $\mu(s)$ is the base measure on the space $\mathcal{S}$.

## 3.2 LVM DROPOUT TRAINING VS. STANDARD DROPOUT TRAINING

Building on the above latent variable model formulation (4) of dropout, we are now ready to point out a simple relation between the training procedure of LVM and that of standard dropout. Given an i.i.d. training sample $D$, the maximum likelihood estimate for the LVM formulation of dropout in (4) is equivalent to minimizing the following negative log-likelihood function:

$$\theta^* = \operatorname*{argmin}_{\theta} -l(D;\theta) = \operatorname*{argmin}_{\theta} -\sum_{i=1}^{N} \log p(y_i|x_i;\theta), \tag{5}$$

where $p(y|x;\theta)$ is given in Eq. (4). Recall the dropout training objective $\mathrm{E}_{S_D}[-l(D,S_D;\theta)]$ in Eq. (2). We have the following theorem as a simple consequence of Jensen's inequality (details in Appendix A):

**Theorem 1.** *The expected loss function of standard dropout (Eq. (2)) is an upper bound of the negative log-likelihood of LVM dropout (Eq. (5)):*

$$-l(D;\theta) \le \mathrm{E}_{S_D}[-l(D,S_D;\theta)]. \tag{6}$$

Theorem 1, in a rigorous sense, justifies dropout training as a convenient and tractable approximation of the LVM formulation in (4). Indeed, since directly minimizing the marginalized negative log-likelihood in (5) may not be easy, a standard practice is to replace the marginalized (conditional) likelihood $p(y|x;\theta)$ in (4) with its empirical Monte carlo average through drawing samples from the dropout variable $S$. The dropout training objective in (2) corresponds exactly to this Monte carlo approximation when a *single* sample $S_i$ is drawn for each training instance $(x_i, y_i)$. Importantly, we note that the above LVM formulation involves only a single network parameter $\theta$, which largely simplifies the picture and facilitates our subsequent analysis.

## 4 EXPECTATION-LINEAR DROPOUT NEURAL NETWORKS

Building on the latent variable model formulation in § 3, we introduce in this section the notion of expectation-linearity that essentially measures the inference gap in (3). We then characterize a general class of neural networks that exhibit expectation-linearity, either exactly or approximately over a distribution $p(x)$ on the input space.

We start with defining expectation-linearity in the simplest single-layer neural network, then we extend the notion into general deep networks in a natural way.

**Definition 1** (Expectation-linear Layer)**.** A network layer $\mathbf{h} = f(x \odot \gamma; \theta)$ is *expectation-linear with respect to* a set $\mathcal{X}' \subseteq \mathcal{X}$, if for all $x \in \mathcal{X}'$ we have

$$\left\| \mathrm{E}[f(x \odot \Gamma; \theta)] - f(x \odot \mathrm{E}[\Gamma]; \theta) \right\|_2 = 0. \tag{7}$$

In this case we say that $\mathcal{X}'$ is *expectation-linearizable*, and $\theta$ is *expectation-linearizing* w.r.t $\mathcal{X}'$.

Obviously, the condition in (7) will guarantee no gap in the dropout inference approximation (3)—an admittedly strong condition that we will relax below. Clearly, if $f$ is an affine function, then we can choose $\mathcal{X}' = \mathcal{X}$ and expectation-linearity is trivial. Note that expectation-linearity depends on the network parameter $\theta$ and the dropout distribution $\Gamma$.

Expectation-linearity, as defined in (7), is overly strong: under standard regularity conditions, essentially the transformation function $f$ has to be affine over the set $\mathcal{X}'$, ruling out for instance the popular sigmoid or tanh activation functions. Moreover, in practice, downstream use of DNNs are usually robust to small errors resulting from *approximate* expectation-linearity (hence the empirical success of dropout), so it makes sense to define an inexact extension. We note also that the definition in (7) is *uniform* over the set $\mathcal{X}'$, while in a statistical setting it is perhaps more meaningful to have expectation-linearity "on average," since inputs from lower density regions are not going to play a significant role anyway. Taking into account the aforementioned motivations, we arrive at the following inexact extension:

**Definition 2** (Approximately Expectation-linear Layer). *A network layer* $\mathbf{h} = f(x \odot \gamma; \theta)$ *is* $\delta$-*approximately expectation-linear with respect to* *a distribution* $p(x)$ *over* $\mathcal{X}$ *if*

$$\mathrm{E}_X \left[ \left\| \mathrm{E}_\Gamma \left[ f(X \odot \Gamma; \theta) | X \right] - f(X \odot \mathrm{E}[\Gamma]; \theta) \right\|_2 \right] < \delta. \tag{8}$$

In this case we say that $p(x)$ is $\delta$-*approximately expectation-linearizable*, and $\theta$ is $\delta$-*approximately expectation-linearizing*.

To appreciate the power of cutting some slack from exact expectation-linearity, we remark that even non-affine activation functions often have approximately linear regions. For example, the logistic function, a commonly used non-linear activation function in DNNs, is approximately linear around the origin. Naturally, we can ask whether it is sufficient for a target distribution $p(x)$ to be well-approximated by an approximately expectation-linearizable one. We begin by providing an appropriate measurement of the quality of this approximation.

**Definition 3** (Closeness, (Andreas et al., 2015)). *A distribution* $p(x)$ *is* $C$-*close to a set* $\mathcal{X}' \subseteq \mathcal{X}$ *if*

$$\mathrm{E} \left[ \inf_{x^* \in \mathcal{X}'} \sup_{\gamma \in \mathcal{S}} \| X \odot \gamma - x^* \odot \gamma \|_2 \right] \le C, \tag{9}$$

where recall that $\mathcal{S}$ is the (bounded) space that the dropout variable lives in.

Intuitively, $p(x)$ is $C$-close to a set $\mathcal{X}'$ if a random sample from $p$ is no more than a distance $C$ from $\mathcal{X}'$ in expectation and under the worst "dropout perturbation". For example, a standard normal distribution is close to an interval centering at origin ($[-\alpha, \alpha]$) with some constant $C$. Our definition of closeness is similar to that in Andreas et al. (2015), who used this notion to analyze self-normalized log-linear models.

We are now ready to state our first major result that quantifies approximate expectation-linearity of a single-layered network (proof in Appendix B.1):

**Theorem 2.** *Given a network layer* $\mathbf{h} = f(x \odot \gamma; \theta)$, *where* $\theta$ *is* expectation-linearizing *w.r.t.* $\mathcal{X}' \subseteq \mathcal{X}$. *Suppose* $p(x)$ *is* $C$-close to $\mathcal{X}'$ *and for all* $x \in \mathcal{X}$, $\| \nabla_x f(x) \|_{\mathsf{op}} \le B$, *where* $\| \cdot \|_{\mathsf{op}}$ *is the usual operator norm. Then,* $p(x)$ *is* $2BC$-*approximately expectation-linearizable.*

Roughly, Theorem 2 states that the input distribution $p(x)$ that place most of its mass on regions close to expectation-linearizable sets are approximately expectation-linearizable on a similar scale. The bounded operator norm assumption on the derivative $\nabla f$ is satisfied in most commonly used layers. For example, for a fully connected layer with weight matrix $W$, bias vector $b$, and activation function $\sigma$, $\| \nabla f(\cdot) \|_{\mathsf{op}} = |\sigma'(\cdot)| \cdot \| W \|_{\mathsf{op}}$ is bounded by $\| W \|_{\mathsf{op}}$ and the supremum of $|\sigma'(\cdot)|$ (1/4 when $\sigma$ is sigmoid and 1 when $\sigma$ is tanh).

Next, we extend the notion of approximate expectation-linearity to deep dropout neural networks.

**Definition 4** (Approximately Expectation-linear Network). *A deep neural network with* $L$ *layers (cf. Eq. (1)) is* $\delta$-*approximately expectation-linear with respect to* $p(x)$ *over* $\mathcal{X}$ *if*

$$\mathrm{E}_X \left[ \left\| \mathrm{E}_S \left[ \mathbf{H}^{(L)}(X, S; \theta) | X \right] - \mathbf{h}^{(L)}(X, \mathrm{E}[S]; \theta) \right\|_2 \right] < \delta. \tag{10}$$

where $\mathbf{h}^{(L)}(X, \mathrm{E}[S]; \theta)$ is the output of the deterministic neural network in standard dropout.

Lastly, we relate the level of approximate expectation-linearity of a deep neural network to the level of approximate expectation-linearity of each of its layers:

**Theorem 3.** *Given an $L$-layer neural network as in Eq. (1), and suppose that each layer $l \in \{1, \ldots, L\}$ is $\delta$-approximately expectation-linear w.r.t. $p(\mathbf{h}^{(l)})$, $\mathrm{E}[\Gamma^{(l)}] \leq \gamma$, $\sup_x \|\nabla f_l(x)\|_{\mathsf{op}} \leq B$, and $\mathrm{E}[\mathrm{Var}[\mathbf{H}^{(l)}|X]] \leq \sigma^2$. Then the network is $\Delta$-approximately expectation-linear with*

$$\Delta = (B\gamma)^{L-1}\delta + (\delta + B\gamma\sigma)\left(\frac{1 - (B\gamma)^{L-1}}{1 - B\gamma}\right). \tag{11}$$

From Theorem 3 (proof in Appendix B.2) we observe that the level of approximate expectation-linearity of the network mainly depends on four factors: the level of approximate expecatation-linearity of each layer ($\delta$), the expected variance of each layer ($\sigma$), the operator norm of the derivative of each layer's transformation function ($B$), and the mean of each layer's dropout variable ($\gamma$). In practice, $\gamma$ is often a constant less than or equal to 1. For example, if $\Gamma \sim \mathrm{Bernoulli}(p)$, then $\gamma = p$.

According to the theorem, the operator norm of the derivative of each layer's transformation function is an important factor in the level of approximate expectation-linearity: the smaller the operator norm is, the better the approximation. Interestingly, the operator norm of a layer often depends on the norm of the layer's weight (e.g. for fully connected layers). Therefore, adding max-norm constraints to regularize dropout neural networks can lead to better approximate expectation-linearity hence smaller inference gap and the often improved model performance.

It should also be noted that when $B\gamma < 1$, the approximation error $\Delta$ tends to be a constant when the network becomes deeper. When $B\gamma = 1$, $\Delta$ grows linearly with $L$, and when $B\gamma > 1$, the growth of $\Delta$ becomes exponential. Thus, it is essential to keep $B\gamma < 1$ to achieve good approximation, particularly for deep neural networks.

## 5 EXPECTATION-LINEAR REGULARIZED DROPOUT

In the previous section we have managed to bound the approximate expectation-linearity, hence the inference gap in (3), of dropout neural networks. In this section, we first prove a uniform deviation bound of the *sampled* approximate expectation-linearity measure from its mean, which motivates adding the sampled (hence computable) expectation-linearity measure as a regularization scheme to standard dropout, with the goal of explicitly controlling the inference gap of the learned parameter, hence potentially improving the performance. Then we give the upper bounds on the loss in accuracy due to expectation-linearization, and describe classes of distributions that expectation-linearize easily.

### 5.1 A UNIFORM DEVIATION BOUND FOR THE SAMPLED EXPECTATION-LINEAR MEASURE

We now show that an expectation-linear network can be found by expectation-linearizing the network on the training sample. To this end, we prove a uniform deviation bound between the empirical expectation-linearization measure using i.i.d. samples (Eq. (12)) and its mean (Eq. (13)).

**Theorem 4.** *Let $\mathcal{H} = \{\mathbf{h}^{(L)}(x, s; \theta) : \theta \in \Theta\}$ denote a space of $L$-layer dropout neural networks indexed with $\theta$, where $\mathbf{h}^{(L)} : \mathcal{X} \times \mathcal{S} \to \mathcal{R}$ and $\Theta$ is the space that $\theta$ lives in. Suppose that the neural networks in $\mathcal{H}$ satisfy the constraints: 1) $\forall x \in \mathcal{X}, \|x\|_2 \leq \alpha$; 2) $\forall l \in \{1, \ldots, L\}, \mathrm{E}(\Gamma^{(l)}) \leq \gamma$ and $\|\nabla f_l\|_{op} \leq B$; 3) $\|\mathbf{h}^{(L)}\| \leq \beta$. Denote empirical expectation-linearization measure and its mean as:*

$$\hat{\Delta} = \frac{1}{n}\sum_{i=1}^{n}\left\|\mathrm{E}_{S_i}\left[\mathbf{H}^{(L)}(X_i, S_i; \theta)\right] - \mathbf{h}^{(L)}(X_i, \mathrm{E}[S_i]; \theta)\right\|_2, \tag{12}$$

$$\Delta = \mathrm{E}_X\left[\left\|\mathrm{E}_S\left[\mathbf{H}^{(L)}(X, S; \theta)\right] - \mathbf{h}^{(L)}(X, \mathrm{E}[S]; \theta)\right\|_2\right]. \tag{13}$$

*Then, with probability at least $1 - \nu$, we have*

$$\sup_{\theta \in \Theta} |\Delta - \hat{\Delta}| < \frac{2\alpha B^L(\gamma^{L/2} + 1)}{\sqrt{n}} + \beta\sqrt{\frac{\log(1/\nu)}{n}}. \tag{14}$$

From Theorem 4 (proof in Appendix C.1) we observe that the deviation bound decreases exponentially with the number of layers $L$ when the operator norm of the derivative of each layer's transformation

function $(B)$ is less than 1 (and the contrary if $B \geq 1$). Importantly, the square root dependence on the number of samples $(n)$ is standard and cannot be improved without significantly stronger assumptions.

It should be noted that Theorem 4 per se does not imply anything between expectation-linearization and the model accuracy (i.e. how well the expectation-linearized neural network actually achieves on modeling the data). Formally studying this relation is provided in § 5.3. In addition, we provide some experimental evidences in § 6 on how improved approximate expectation-linearity (equivalently smaller inference gap) does lead to better empirical performances.

## 5.2 EXPECTATION-LINEARIZATION AS REGULARIZATION

The uniform deviation bound in Theorem 4 motivates the possibility of obtaining an approximately expectation-linear dropout neural networks through adding the empirical measure (12) as a regularization scheme to the standard dropout training objective, as follows:

$$loss(D; \theta) = -l(D; \theta) + \lambda V(D; \theta), \tag{15}$$

where $-l(D; \theta)$ is the negative log-likelihood defined in Eq. (5), $\lambda > 0$ is a regularization constant, and $V(D; \theta)$ measures the level of approximate expectation-linearity:

$$V(D; \theta) = \frac{1}{N} \sum_{i=1}^{N} \left\| \mathrm{E}_{S_i} \left[ \mathbf{H}^{(L)}(x_i, S_i; \theta) \right] - \mathbf{h}^{(L)}(x_i, \mathrm{E}[S_i]; \theta) \right\|_2^2. \tag{16}$$

To solve (15), we can minimize $loss(D; \theta)$ via stochastic gradient descent as in standard dropout, and approximate $V(D; \theta)$ using Monte carlo:

$$V(D; \theta) \approx \frac{1}{N} \sum_{i=1}^{N} \left\| \mathbf{h}^{(L)}(x_i, s_i; \theta) - \mathbf{h}^{(L)}(x_i, \mathrm{E}[S_i]; \theta) \right\|_2^2, \tag{17}$$

where $s_i$ is the same dropout sample as in $l(D; \theta)$ for each training instance in a mini-batch. Thus, the only additional computational cost comes from the deterministic term $\mathbf{h}^{(L)}(x_i, \mathrm{E}[S_i]; \theta)$. Overall, our regularized dropout (15), in its Monte carlo approximate form, is as simple and efficient as the standard dropout.

## 5.3 ON THE ACCURACY OF EXPECTATION-LINEARIZED MODELS

So far our discussion has concentrated on the problem of finding expectation-linear neural network models, without any concerns on how well they actually perform at modeling the data. In this section, we characterize the trade-off between maximizing "data likelihood" and satisfying an expectation-linearization constraint.

To achieve the characterization, we measure the *likelihood gap* between the classical maximum likelihood estimator (MLE) and the MLE subject to a expectation-linearization constraint. Formally, given training data $D = \{(x_1, y_1), \ldots, (x_n, y_n)\}$, we define

$$\hat{\theta} = \underset{\theta \in \Theta}{\mathrm{argmin}} \quad -l(D; \theta) \tag{18}$$

$$\hat{\theta}_\delta = \underset{\theta \in \Theta, V(D; \theta) \leq \delta}{\mathrm{argmin}} -l(D; \theta) \tag{19}$$

where $-l(D; \theta)$ is the negative log-likelihood defined in Eq. (5), and $V(D; \theta)$ is the level of approximate expectation-linearity in Eq. (16).

We would like to control the loss of model accuracy by obtaining a bound on the *likelihood gap* defined as:

$$\Delta_l(\hat{\theta}, \hat{\theta}_\delta) = \frac{1}{n}(l(D; \hat{\theta}) - l(D; \hat{\theta}_\delta)) \tag{20}$$

In the following, we focus on neural networks with *softmax* output layer for classification tasks.

$$p(y|x, s; \theta) = \mathbf{h}_y^{(L)}(x, s; \theta) = f_L(\mathbf{h}^{(L-1)}(x, s); \eta) = \frac{e^{\eta_y^T \mathbf{h}^{(L-1)}(x,s)}}{\sum\limits_{y' \in \mathcal{Y}} e^{\eta_{y'}^T \mathbf{h}^{(L-1)}(x,s)}} \tag{21}$$

where $\theta = \{\theta_1, \ldots, \theta_{L-1}, \eta\}$, $\mathcal{Y} = \{1, \ldots, k\}$ and $\eta = \{\eta_y : y \in \mathcal{Y}\}$. We claim:

**Theorem 5.** *Given an $L$-layer neural network $\mathbf{h}^{(L)}(x, s; \theta)$ with softmax output layer in (21), where parameter $\theta \in \Theta$, dropout variable $s \in \mathcal{S}$, input $x \in \mathcal{X}$ and target $y \in \mathcal{Y}$. Suppose that for every $x$ and $s$, $p(y|x, s; \hat{\theta})$ makes a unique best prediction—that is, for each $x \in \mathcal{X}, s \in \mathcal{S}$, there exists a unique $y^* \in \mathcal{Y}$ such that $\forall y \neq y^*$, $\hat{\eta}_y^T \mathbf{h}^{(L-1)}(x, s) < \hat{\eta}_{y^*}^T \mathbf{h}^{(L-1)}(x, s)$. Suppose additionally that $\forall x, s, \|\mathbf{h}^{(L-1)}(x, s; \hat{\theta})\| \leq \beta$, and $\forall y, p(y|x; \hat{\theta}) > 0$. Then*

$$\Delta_l(\hat{\theta}, \hat{\theta}_\delta) \leq c_1 \beta^2 \left( \|\hat{\eta}\|_2 - \frac{\delta}{4\beta} \right)^2 e^{-c_2 \delta / 4\beta} \tag{22}$$

*where $c_1$ and $c_2$ are distribution-dependent constants.*

From Theorem 5 (proof in Appendix C.2) we observe that, at one extreme, distributions closed to deterministic can be expectation-linearized with little loss of likelihood.

What about the other extreme — distributions "as close to uniform distribution as possible"? With suitable assumptions about the form of $p(y|x, s; \hat{\theta})$ and $p(y|x; \hat{\theta})$, we can achieve an accuracy loss bound for distributions that are close to uniform:

**Theorem 6.** *Suppose that $\forall x, s, \|\mathbf{h}^{(L-1)}(x, s; \hat{\theta})\| \leq \beta$. Additionally, for each $(x_i, y_i) \in D, s \in \mathcal{S}$, $\log \frac{1}{k} \leq \log p(y_i|x_i, s; \hat{\theta}) \leq \frac{1}{k} \sum_{y \in \mathcal{Y}} \log p(y|x_i, s; \hat{\theta})$. Then asymptotically as $n \to \infty$:*

$$\Delta_l(\hat{\theta}, \hat{\theta}_\delta) \leq \left( 1 - \frac{\delta}{4\beta \|\hat{\eta}\|_2} \right) \mathrm{E} \left[ \mathrm{KL} \left( p(\cdot|X; \theta) \| \mathrm{Unif}(\mathcal{Y}) \right) \right] \tag{23}$$

Theorem 6 (proof in Appendix C.3) indicates that uniform distributions are also an easy class for expectation-linearization.

The next question is whether there exist any classes of conditional distributions $p(y|x)$ for which all distributions are provably hard to expectation-linearize. It remains an open problem and might be an interesting direction for future work.

## 6 EXPERIMENTS

In this section, we evaluate the empirical performance of the proposed regularized dropout in (15) on a variety of network architectures for the classification task on three benchmark datasets—MNIST, CIFAR-10 and CIFAR-100. We applied the same data preprocessing procedure as in Srivastava et al. (2014). To make a thorough comparison and provide experimental evidence on how the expectation-linearization interacts with the predictive power of the learned model, we perform experiments of Monte Carlo (MC) dropout, which approximately computes the final prediction (left-hand side of (3)) via Monte Carlo sampling, w/o the proposed regularizer. In the case of MC dropout, we average $m = 100$ predictions using randomly sampled configurations. In addition, the network architectures and hyper-parameters for each experiment setup are the same as those in Srivastava et al. (2014), unless we explicitly claim to use different ones. Following previous works, for each data set We held out 10,000 random training images for validation to tune the hyper-parameters, including $\lambda$ in Eq. (15). When the hyper-parameters are fixed, we train the final models with all the training data, including the validation data. A more detailed description of the conducted experiments can be provided in Appendix D. For each experiment, we report the mean test errors with corresponding standard deviations over 5 repetitions.

### 6.1 MNIST

The MNIST dataset (LeCun et al., 1998) consists of 70,000 handwritten digit images of size $28 \times 28$, where 60,000 images are used for training and the rest for testing. This task is to classify the images into 10 digit classes. For the purpose of comparison, we train 6 neural networks with different architectures. The experimental results are shown in Table 1.

### 6.2 CIFAR-10 AND CIFAR-100

The CIFAR-10 and CIFAR-100 datasets (Krizhevsky, 2009) consist of 60,000 color images of size $32 \times 32$, drawn from 10 and 100 categories, respectively. 50,000 images are used for training and the

Table 1: Comparison of classification error percentage on test data with and without using expectation-linearization on MNIST, CIFAR-10 and CIFAR-100, under different network architectures (with standard deviations for 5 repetitions).

| Data | Architecture | w.o. EL | | w. EL | |
|---|---|---|---|---|---|
| | | Standard | MC | Standard | MC |
| MNIST | 3 dense,1024,logistic | 1.23±0.03 | 1.06±0.02 | 1.07±0.02 | 1.06±0.03 |
| | 3 dense,1024,relu | 1.19±0.02 | 1.04±0.02 | 1.03±0.02 | 1.05±0.03 |
| | 3 dense,1024,relu+max-norm | 1.05±0.03 | 1.02±0.02 | 0.98±0.03 | 1.02±0.02 |
| | 3 dense,2048,relu+max-norm | 1.07±0.02 | 1.00±0.02 | 0.94±0.02 | 0.97±0.03 |
| | 2 dense,4096,relu+max-norm | 1.03±0.02 | 0.92±0.03 | 0.90±0.02 | 0.93±0.02 |
| | 2 dense,8192,relu+max-norm | 0.99±0.02 | 0.96±0.02 | 0.87±0.02 | 0.92±0.03 |
| CIFAR-10 | 3 conv+2 dense,relu+max-norm | 12.82±0.10 | 12.16±0.12 | 12.20±0.14 | 12.21±0.15 |
| CIFAR-100 | 3 conv+2 dense,relu+max-norm | 37.22±0.22 | 36.01±0.21 | 36.25±0.12 | 36.10±0.18 |

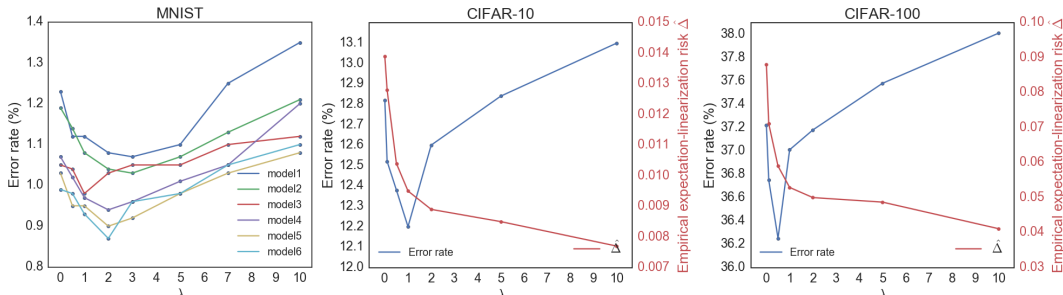

Figure 1: Error rate and empirical expectation-linearization risk relative to $\lambda$.

rest for testing. The neural network architecture we used for these two datasets has 3 convolutional layers, followed by two fully-connected (dense) hidden layers (again, same as that in Srivastava et al. (2014)). The experimental results are recorded in Table 1, too.

From Table 1 we can see that on MNIST data, dropout network training with expectation-linearization outperforms standard dropout on all 6 neural architectures. On CIFAR data, expectation-linearization reduces error rate from 12.82% to 12.20% for CIFAR-10, achieving 0.62% improvement. For CIFAR-100, the improvement in terms of error rate is 0.97% with reduction from 37.22% to 36.25%.

From the results we see that with or without expectation-linearization, the MC dropout networks achieve similar results. It illustrates that by achieving expectation-linear neural networks, the predictive power of the learned models has not degraded significantly. Moreover, it is interesting to see that with the regularization, on MNIST dataset, standard dropout networks achieve even better accuracy than MC dropout. It may be because that with expectation-linearization, standard dropout inference achieves better approximation of the final prediction than MC dropout with (only) 100 samples. On CIFAR datasets, MC dropout networks achieve better accuracy than the ones with the regularization. But, obviously, MC dropout requires much more inference time than standard dropout (MC dropout with $m$ samples requires about $m$ times the inference time of standard dropout).

## 6.3 EFFECT OF REGULARIZATION CONSTANT $\lambda$

In this section, we explore the effect of varying the hyper-parameter for the expectation-linearization rate $\lambda$. We train the network architectures in Table 1 with the $\lambda$ value ranging from 0.1 to 10.0. Figure 1 shows the test errors obtained as a function of $\lambda$ on three datasets. In addition, Figure 1, middle and right panels, also measures the empirical expectation-linearization risk $\hat{\Delta}$ of Eq. (12) with varying $\lambda$ on CIFAR-10 and CIFAR-100, where $\hat{\Delta}$ is computed using Monte carlo with 100 independent samples.

From Figure 1 we can see that when $\lambda$ increases, better expectation-linearity is achieved (i.e. $\hat{\Delta}$ decreases). The model accuracy, however, has not kept growing with increasing $\lambda$, showing that in practice considerations on the trade-off between model expectation-linearity and accuracy are needed.

Table 2: Comparison of test data errors using standard dropout, Monte Carlo dropout, standard dropout with our proposed expectation-linearization, and recently proposed dropout distillation on CIFAR-10 and CIFAR-100 under AllConv, (with standard deviations for 5 repetitions).

| Data | Network | Standard | MC | w. EL | Distillation |
|---|---|---|---|---|---|
| CIFAR-10 | AllConv | 11.18±0.11 | 10.58±0.21 | 10.86±0.08 | 10.81±0.14 |
| CIFAR-100 | AllConv | 35.50±0.23 | 34.43±0.25 | 35.10±0.13 | 35.07±0.20 |

### 6.4 COMPARISON WITH DROPOUT DISTILLATION

To make a thorough empirical comparison with the recently proposed Dropout Distillation method (Bulò et al., 2016), we also evaluate our regularization method on CIFAR-10 and CIFAR-100 datasets with the All Convolutional Network (Springenberg et al., 2014) (AllConv). To facilitate comparison, we adopt the originally reported hyper-parameters and the same setup for training.

Table 2 gives the results comparison the classification error percentages on test data under AllConv using standard dropout, Monte Carlo dropout, standard dropout with our proposed expectation-linearization, and recently proposed dropout distillation on CIFAR-10 and CIFAR-100 [1]. According to Table 2, our proposed expectation-linear regularization method achieves comparable performance to dropout distillation.

## 7 CONCLUSIONS

In this work, we attempted to establish a theoretical basis for the understanding of dropout, motivated by controlling the gap between dropout's training and inference phases. Through formulating dropout as a latent variable model and introducing the notion of (approximate) expectation-linearity, we have formally studied the inference gap of dropout, and introduced an empirical measure as a regularization scheme to explicitly control the gap. Experiments on three benchmark datasets demonstrate that reducing the inference gap can indeed improve the end performance. In the future, we intend to formally relate the inference gap to the generalization error of the underlying network, hence providing further justification of regularized dropout.

### ACKNOWLEDGEMENTS

This research was supported in part by DARPA grant FA8750-12-2-0342 funded under the DEFT program. Any opinions, findings, and conclusions or recommendations expressed in this material are those of the authors and do not necessarily reflect the views of DARPA.

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

APPENDIX: DROPOUT WITH EXPECTATION-LINEAR REGULARIZATION

## A LVM DROPOUT TRAINING VS. STANDARD DROPOUT TRAINING

**Proof of Theorem 1**

*Proof.*

$$
\begin{aligned}
\mathrm{E}_{S_D}[l(D, S_D; \theta)] &= \int_{\mathcal{S}} \prod_{i=1}^{N} p(s_i) \Big( \sum_{i=1}^{N} \log p(y_i|x_i, s_i; \theta) \Big) d\mu(s_1) \dots d\mu(s_N) \\
&= \sum_{i=1}^{N} \int_{\mathcal{S}} p(s_i) \log p(y_i|x_i, s_i; \theta) d\mu(s_i)
\end{aligned}
$$

Because $\log(\cdot)$ is a concave function, from Jensen's Inequality,

$$
\int_{\mathcal{S}} p(s) \log p(y|x, s; \theta) d\mu(s) \leq \log \int_{\mathcal{S}} p(s) p(y|x, s; \theta) d\mu(s)
$$

Thus

$$
\mathrm{E}_{S_D}[-l(D, S_D; \theta)] \geq \sum_{i=1}^{N} \log \int_{\mathcal{S}} p(s_i) p(y_i|x_i, s_i; \theta) d\mu(s_i) = -l(D; \theta).
$$

$\square$

## B EXPECTATION-LINEAR DROPOUT NEURAL NETWORKS

### B.1 PROOF OF THEOREM 2

*Proof.* Let $\gamma^* = \mathrm{E}[\Gamma]$, and

$$
A \triangleq \{x : \|\mathrm{E}[f(x \odot \Gamma; \theta)] - f(x \odot \gamma^*; \theta)\|_2 = 0\}
$$

Let $X^* = \operatorname*{argmin}_{x \in A} \sup_{\gamma \in \mathcal{S}} \|X \odot \gamma - x \odot \gamma\|_2$, and $X^- = X - X^*$. Then,

$$
X \odot \gamma = X^* \odot \gamma + X^- \odot \gamma
$$

In the following, we omit the parameter $\theta$ for convenience. Moreover, we denote

$$
\mathrm{E}_{\Gamma}[f(X \odot \Gamma; \theta)] \triangleq \mathrm{E}[f(X \odot \Gamma; \theta)|X]
$$

From Taylor Series, there exit some $X', X'' \in \mathcal{X}$ satisfy that

$$
\begin{aligned}
f(X \odot \Gamma) &= f(X^* \odot \Gamma) + f'(X' \odot \Gamma)(X^- \odot \Gamma) \\
f(X \odot \gamma^*) &= f(X^* \odot \gamma^*) + f'(X'' \odot \gamma^*)(X^- \odot \gamma^*)
\end{aligned}
$$

where we denote $f'(x) = (\nabla_x f(x))^T$. Then,

$$
\begin{aligned}
& \mathrm{E}_{\Gamma}[f(X \odot \Gamma) - f(X \odot \gamma^*)] \\
=~& \mathrm{E}_{\Gamma}[f(X^* \odot \Gamma + X^- \odot \Gamma) - f(X^* \odot \gamma^* + X^- \odot \gamma^*)] \\
=~& \mathrm{E}_{\Gamma}[f(X^* \odot \Gamma) - f(X^* \odot \gamma^*) + f'(X' \odot \Gamma)(X^- \odot \Gamma) - f'(X'' \odot \gamma^*)(X^- \odot \gamma^*)] \\
=~& \mathrm{E}_{\Gamma}[f(X^* \odot \Gamma) - f(X^* \odot \gamma^*)] + \mathrm{E}_{\Gamma}[f'(X' \odot \Gamma)(X^- \odot \Gamma) - f'(X'' \odot \gamma^*)(X^- \odot \gamma^*)]
\end{aligned}
$$

Since $X^* \in A$, we have

$$
\mathrm{E}_{\Gamma}[f(X^* \odot \Gamma) - f(X^* \odot \gamma^*)] = 0.
$$

Then,

$$
\begin{aligned}
& \mathrm{E}_{\Gamma}[f(X \odot \Gamma) - f(X \odot \gamma^*)] \\
=~& \mathrm{E}_{\Gamma}[f'(X' \odot \Gamma)(X^- \odot \Gamma) - f'(X'' \odot \gamma^*)(X^- \odot \gamma^*)] \\
=~& \mathrm{E}_{\Gamma}[(f'(X' \odot \Gamma) - f'(X'' \odot \gamma^*))(X^- \odot \Gamma)] + \mathrm{E}_{\Gamma}[f'(X'' \odot \gamma^*)(X^- \odot \Gamma - X^- \odot \gamma^*)] \\
=~& \mathrm{E}_{\Gamma}[(f'(X' \odot \Gamma) - f'(X'' \odot \gamma^*))(X^- \odot \Gamma)]
\end{aligned}
$$

Then,
$$\|\mathrm{E}_\Gamma[f(X \odot \Gamma)] - f(X \odot \gamma^*)\|_2$$
$$= \|\mathrm{E}_\Gamma[(f'(X' \odot \Gamma) - f'(X'' \odot \gamma^*))(X^- \odot \Gamma)]\|_2$$

Since $\|X^- \odot \gamma'\|_2 \leq \sup_{\gamma \in \mathcal{S}} \|X^- \odot \gamma\|_2 = \inf_{x \in A} \sup_{\gamma \in \mathcal{S}} \|X \odot \gamma - x \odot \gamma\|_2$, and from Jensen's inequality and property of operator norm,

$$\|\mathrm{E}_\Gamma[(f'(X' \odot \Gamma) - f'(X'' \odot \gamma^*))(X^- \odot \Gamma)]\|_2$$
$$\leq \mathrm{E}_\Gamma\left[\|f'(X' \odot \Gamma) - f'(X'' \odot \gamma^*)\|_{op}\|X^- \odot \Gamma\|_2\right]$$
$$\leq 2B\mathrm{E}_\Gamma\left[\|X^- \odot \Gamma\|_2\right]$$
$$\leq 2B \inf_{x \in A} \sup_{\gamma \in \mathcal{S}} \|X \odot \gamma - x \odot \gamma\|_2$$

Finally we have,

$$\mathrm{E}_X\left[\|\mathrm{E}_\Gamma[(f'(X' \odot \Gamma) - f'(X'' \odot \gamma^*))(X^- \odot \Gamma)]\|_2\right]$$
$$\leq 2B\mathrm{E}\left[\inf_{x \in A} \sup_{\gamma \in \mathcal{S}} \|X \odot \gamma - x \odot \gamma\|_2\right] \leq 2BC$$

$\square$

### B.2 PROOF OF THEOREM 3

*Proof.* Induction on the number of the layers $L$. As before, we omit the parameter $\theta$.
**Initial step:** when $L = 1$, the statement is obviously true.
**Induction on $L$:** Suppose that the statement is true for neural networks with $L$ layers.
Now we prove the case $L + 1$. From the inductive assumption, we have,

$$\mathrm{E}_X\left[\|\mathrm{E}_{S_L}[\mathbf{H}^{(L)}(X, S_L)] - \mathbf{h}^{(L)}(X, \mathrm{E}[S_L])\|_2\right] \leq \Delta_L \tag{1}$$

where $S_L = \{\Gamma^{(1)}, \ldots, \Gamma^{(L)}\}$ is the dropout random variables for the first $L$ layers, and

$$\Delta_L = (B\gamma)^{L-1}\delta + (\delta + B\gamma\sigma)\left(\frac{1 - (B\gamma)^{L-1}}{1 - B\gamma}\right)$$

In addition, the $L + 1$ layer is $\delta$-approximately expectation-linear, we have:

$$\mathrm{E}_{\mathbf{H}^{(L)}}\left[\|\mathrm{E}_{\Gamma^{(L+1)}}[f_{L+1}(\mathbf{H}^{(L)} \odot \Gamma^{(L+1)})] - f_{L+1}(\mathbf{H}^{(L)} \odot \gamma^{(L+1)})\|_2\right] \leq \delta \tag{2}$$

Let $\mathrm{E}[\Gamma^{(l)}] = \gamma^{(l)}, \forall l \in \{1, \ldots, L+1\}$, and let $\mathbf{H}^{(l)}$ and $\mathbf{h}^{(l)}$ be short for $\mathbf{H}^{(l)}(X, S_l)$ and $\mathbf{h}^{(l)}(X, \mathrm{E}(S_l))$, respectively, when there is no ambiguity. Moreover, we denote

$$\mathrm{E}_S[\mathbf{H}^{(L)}(X, S; \theta)] = \mathrm{E}_S[\mathbf{H}^{(L)}(X, S; \theta)|X]$$

for convenience. Then,

$$\mathrm{E}_X\left[\|\mathrm{E}_{S_{L+1}}[\mathbf{H}^{(L+1)}] - \mathbf{h}^{(L+1)}\|_2\right]$$
$$= \mathrm{E}_X\left[\left\|\mathrm{E}_{S_L}\left[\mathrm{E}_{\Gamma^{(L+1)}}[f_{L+1}(\mathbf{H}^{(L)} \odot \Gamma^{(L+1)})] - f_{L+1}(\mathbf{h}^{(L)} \odot \gamma^{(L+1)})\right]\right.\right.$$
$$\left.\left.+ \mathrm{E}_{S_L}\left[f_{L+1}(\mathbf{H}^{(L)} \odot \gamma^{(L+1)})\right] - f_{L+1}(\mathbf{h}^{(L)} \odot \gamma^{(L+1)})\right\|_2\right]$$
$$\leq \mathrm{E}_X\left[\left\|\mathrm{E}_{S_L}\left[\mathrm{E}_{\Gamma^{(L+1)}}[f_{L+1}(\mathbf{H}^{(L)} \odot \Gamma^{(L+1)})] - f_{L+1}(\mathbf{h}^{(L)} \odot \gamma^{(L+1)})\right]\right\|_2\right]$$
$$+ \mathrm{E}_X\left[\left\|\mathrm{E}_{S_L}\left[f_{L+1}(\mathbf{H}^{(L)} \odot \gamma^{(L+1)})\right] - f_{L+1}(\mathbf{h}^{(L)} \odot \gamma^{(L+1)})\right\|_2\right]$$

From Eq. 2 and Jensen's inequality, we have

$$\mathrm{E}_X\left[\left\|\mathrm{E}_{S_L}\left[\mathrm{E}_{\Gamma^{(L+1)}}[f_{L+1}(\mathbf{H}^{(L)} \odot \Gamma^{(L+1)})] - f_{L+1}(\mathbf{h}^{(L)} \odot \gamma^{(L+1)})\right]\right\|_2\right]$$
$$\leq \mathrm{E}_{\mathbf{H}^{(L)}}\left[\left\|\mathrm{E}_{\Gamma^{(L+1)}}[f_{L+1}(\mathbf{H}^{(L)} \odot \Gamma^{(L+1)})] - f_{L+1}(\mathbf{h}^{(L)} \odot \gamma^{(L+1)})\right\|_2\right] \leq \delta \tag{3}$$

and

$$\mathrm{E}_X\left[\left\|\mathrm{E}_{S_L}\left[f_{L+1}(\mathbf{H}^{(L)}\odot\gamma^{(L+1)})\right]-f_{L+1}(\mathbf{h}^{(L)}\odot\gamma^{(L+1)})\right\|_2\right]$$

$$= \mathrm{E}_X\left[\left\|\mathrm{E}_{S_L}\left[f_{L+1}(\mathbf{H}^{(L)}\odot\gamma^{(L+1)})\right]-f_{L+1}(\mathrm{E}_{S_L}\left[\mathbf{H}^{(L)}\right]\odot\gamma^{(L+1)})\right.\right.$$
$$\left.\left.+f_{L+1}(\mathrm{E}_{S_L}\left[\mathbf{H}^{(L)}\right]\odot\gamma^{(L+1)})-f_{L+1}(\mathbf{h}^{(L)}\odot\gamma^{(L+1)})\right\|_2\right] \tag{4}$$

$$\leq \mathrm{E}_X\left[\left\|\mathrm{E}_{S_L}\left[f_{L+1}(\mathbf{H}^{(L)}\odot\gamma^{(L+1)})\right]-f_{L+1}(\mathrm{E}_{S_L}\left[\mathbf{H}^{(L)}\right]\odot\gamma^{(L+1)})\right\|_2\right]$$
$$+\mathrm{E}_X\left[\left\|f_{L+1}(\mathrm{E}_{S_L}\left[\mathbf{H}^{(L)}\right]\odot\gamma^{(L+1)})-f_{L+1}(\mathbf{h}^{(L)}\odot\gamma^{(L+1)})\right\|_2\right]$$

Using Jensen's inequality, property of operator norm and $\mathrm{E}\left[\mathrm{Var}[\mathbf{H}^{(l)}|X]\right]\leq\sigma^2$, we have

$$\mathrm{E}_X\left[\left\|\mathrm{E}_{S_L}\left[f_{L+1}(\mathbf{H}^{(L)}\odot\gamma^{(L+1)})\right]-f_{L+1}(\mathrm{E}_{S_L}\left[\mathbf{H}^{(L)}\right]\odot\gamma^{(L+1)})\right\|_2\right]$$

$$\leq \mathrm{E}_{\mathbf{H}^{(L)}}\left[\left\|f_{L+1}(\mathbf{H}^{(L)}\odot\gamma^{(L+1)})-f_{L+1}(\mathrm{E}_{S_L}\left[\mathbf{H}^{(L)}\right]\odot\gamma^{(L+1)})\right\|_2\right] \tag{5}$$

$$\leq B\gamma\mathrm{E}_{\mathbf{H}^{(L)}}\left[\left\|\mathbf{H}^{(L)}-\mathrm{E}_{S_L}\left[\mathbf{H}^{(L)}\right]\right\|_2\right]$$

$$\leq B\gamma\left(\mathrm{E}_{\mathbf{H}^{(L)}}\left[\left\|\mathbf{H}^{(L)}-\mathrm{E}_{S_L}\left[\mathbf{H}^{(L)}\right]\right\|_2^2\right]\right)^{\frac{1}{2}}\leq B\gamma\sigma$$

From Eq. 1

$$\mathrm{E}_X\left[\left\|f_{L+1}(\mathrm{E}_{S_L}\left[\mathbf{H}^{(L)}\right]\odot\gamma^{(L+1)})-f_{L+1}(\mathbf{h}^{(L)}\odot\gamma^{(L+1)})\right\|_2\right]$$

$$= B\gamma\mathrm{E}_X\left[\left\|\mathrm{E}_{S_L}\left[\mathbf{H}^{(L)}\right]-\mathbf{h}^{(L)}\right\|_2\right]\leq B\gamma\Delta_L \tag{6}$$

Finally, to sum up with Eq. 3, Eq. 4, , Eq. 5, , Eq. 6, we have

$$\mathrm{E}_X\left[\left\|\mathrm{E}_{S_{L+1}}\left[\mathbf{H}^{(L+1)}\right]-\mathbf{h}^{(L+1)}\right\|_2\right]$$
$$\leq \delta+B\gamma\sigma+B\gamma\Delta_L$$
$$= (B\gamma)^L\delta+(\delta+B\gamma\sigma)\left(\frac{1-(B\gamma)^L}{1-B\gamma}\right)=\Delta_{L+1}$$

$\square$

## C  EXPECTATION-LINEARIZATION

### C.1  PROOF OF THEOREM 4: UNIFORM DEVIATION BOUND

Before proving Theorem 4, we first define the notations.

Let $X^n=\{X_1,\ldots,X_n\}$ be a set of $n$ samples of input $X$. For a function space $\mathcal{F}:\mathcal{X}\to\mathcal{R}$, we use $Rad_n(\mathcal{F},X^n)$ to denote the *empirical Rademacher complexity* of $\mathcal{F}$,

$$Rad_n(\mathcal{F},X^n)=\mathrm{E}_\sigma\left[\sup_{f\in\mathcal{F}}\left(\frac{1}{n}\sum_{i=1}^n\sigma_if(X_i)\right)\right]$$

and the *Rademacher complexity* is defined as

$$Rad_n(\mathcal{F})=\mathrm{E}_{X^n}\left[Rad_n(\mathcal{F},X^n)\right]$$

In addition, we import the definition of *dropout Rademacher complexity* from Gao & Zhou (2014):

$$\mathcal{R}_n(\mathcal{H},X^n,S^n) = \mathrm{E}_\sigma\left[\sup_{h\in\mathcal{H}}\left(\frac{1}{n}\sum_{i=1}^n\sigma_ih(X_i,S_i)\right)\right]$$
$$\mathcal{R}_n(\mathcal{H}) = \mathrm{E}_{X^n,S^n}\left[Rad_n(\mathcal{H},X^n,S^n)\right]$$

where $\mathcal{H} : \mathcal{X} \times \mathcal{S} \to \mathcal{R}$ is a function space defined on input space $\mathcal{X}$ and dropout variable space $\mathcal{S}$. $\mathcal{R}_n(\mathcal{H}, X^n, S^n)$ and $\mathcal{R}_n(\mathcal{H})$ are the empirical dropout Rademacher complexity and dropout Rademacher complexity, respectively. We further denote $\mathcal{R}_n(\mathcal{H}, X^n) \triangleq \mathrm{E}_{S^n}\Big[ Rad_n(\mathcal{H}, X^n, S^n) \Big]$.

Now, we define the following function spaces:

$$
\begin{aligned}
\mathcal{F} &= \left\{ f(x; \theta) : f(x; \theta) = \mathrm{E}_S\Big[\mathbf{H}^{(L)}(x, S; \theta)\Big], \theta \in \Theta \right\} \\
\mathcal{G} &= \left\{ g(x; \theta) : g(x; \theta) = \mathbf{h}^{(L)}(x, \mathrm{E}[S]; \theta), \theta \in \Theta \right\} \\
\mathcal{H} &= \left\{ h(x, s; \theta) : h(x, s; \theta) = \mathbf{h}^{(L)}(x, s; \theta), \theta \in \Theta \right\}
\end{aligned}
$$

Then, the function space of $v(x) = f(x) - g(x)$ is $\mathcal{V} = \{ f(x) - g(x) : f \in \mathcal{F}, g \in \mathcal{G} \}$.

**Lemma 7.**
$$
Rad_n(\mathcal{F}, X^n) \leq \mathcal{R}_n(\mathcal{H}, X^n)
$$

*Proof.*

$$
\begin{aligned}
\mathcal{R}_n(\mathcal{H}, X^n) &= \mathrm{E}_{S^n}\Big[ Rad_n(\mathcal{H}, X^n, S^n) \Big] \\
&= \mathrm{E}_{S^n}\Big[ \mathrm{E}_\sigma \Big[ \sup_{h \in \mathcal{H}} \Big( \tfrac{1}{n} \sum_{i=1}^n \sigma_i h(X_i, S_i) \Big) \Big] \Big] \\
&= \mathrm{E}_\sigma \Big[ \mathrm{E}_{S^n} \Big[ \sup_{h \in \mathcal{H}} \Big( \tfrac{1}{n} \sum_{i=1}^n \sigma_i h(X_i, S_i) \Big) \Big] \Big] \\
&\geq \mathrm{E}_\sigma \Big[ \sup_{h \in \mathcal{H}} \mathrm{E}_{S^n} \Big[ \Big( \tfrac{1}{n} \sum_{i=1}^n \sigma_i h(X_i, S_i) \Big) \Big] \Big] \\
&= \mathrm{E}_\sigma \Big[ \sup_{h \in \mathcal{H}} \Big( \tfrac{1}{n} \sum_{i=1}^n \sigma_i \mathrm{E}_{S_i} \big[ h(X_i, S_i) \big] \Big) \Big] \\
&= \mathrm{E}_\sigma \Big[ \sup_{h \in \mathcal{H}} \Big( \tfrac{1}{n} \sum_{i=1}^n \sigma_i \mathrm{E}_{S_i} \big[ \mathbf{H}^{(L)}(X_i, S_i; \theta) \big] \Big) \Big] = Rad_n(\mathcal{F}, X^n)
\end{aligned}
$$

$\square$

From Lemma 7, we have $Rad_n(\mathcal{F}) \leq \mathcal{R}_n(\mathcal{H})$.

**Lemma 8.**
$$
\begin{aligned}
\mathcal{R}_n(\mathcal{H}) &\leq \frac{\alpha B^L \gamma^{L/2}}{\sqrt{n}} \\
Rad_n(\mathcal{G}) &\leq \frac{\alpha B^L}{\sqrt{n}}
\end{aligned}
$$

*Proof.* See Theorem 4 in Gao & Zhou (2014). $\square$

Now, we can prove Theorem 4.

**Proof of Theorem 4**

*Proof.* From Rademacher-based uniform bounds theorem, with probability $\geq 1 - \delta$,

$$
\sup_{v \in \mathcal{V}} |\Delta - \hat{\Delta}| < 2 Rad_n(\mathcal{V}) + \beta \sqrt{\frac{\log(1/\delta)}{n}}
$$

Since $\mathcal{V} = \mathcal{F} - \mathcal{G}$, we have

$$
Rad_n(\mathcal{V}) = Rad_n(\mathcal{F} - \mathcal{G}) \leq Rad_n(\mathcal{F}) + Rad_n(\mathcal{G}) \leq \frac{\alpha B^L (\gamma^{L/2} + 1)}{\sqrt{n}}
$$

Then, finally, we have that with probability $\geq 1 - \delta$,

$$
\sup_{\theta \in \Theta} |\Delta - \hat{\Delta}| < \frac{2 \alpha B^L (\gamma^{L/2} + 1)}{\sqrt{n}} + \beta \sqrt{\frac{\log(1/\delta)}{n}}
$$

$\square$

## C.2 PROOF OF THEOREM 5: NON-UNIFORM BOUND OF MODEL ACCURACY

For convenience, we denote $\lambda = \{\theta_1, \dots, \theta_{L-1}\}$. Then $\theta = \{\lambda, \eta\}$, and MLE $\hat{\theta} = \{\hat{\lambda}, \hat{\eta}\}$

**Lemma 9.**
$$\|\nabla f_L(\cdot; \eta)^T\|_{op} \leq 2\|\eta\|_2 \tag{7}$$

*Proof.* denote
$$A = \nabla f_L(\cdot; \eta)^T = \left[ p_y(\eta_y - \overline{\eta})^T \right]\Big|_{y=1}^{k}$$

where $p_y = p(y|x, s; \theta)$, $\overline{\eta} = \mathrm{E}\left[\eta_Y\right] = \sum\limits_{y=1}^{k} p_y \eta_y$.

For each $v$ such that $\|v\|_2 = 1$,

$$
\begin{aligned}
\|Av\|_2^2 &= \sum_{y \in \mathcal{Y}} \left( p_y \left(\eta_y - \overline{\eta}\right)^T v \right)^2 \leq \sum_{y \in \mathcal{Y}} \|p_y \left(\eta_y - \overline{\eta}\right)\|_2^2 \|v\|_2^2 = \sum_{y \in \mathcal{Y}} \|p_y \left(\eta_y - \overline{\eta}\right)\|_2^2 \\
&\leq \sum_{y \in \mathcal{Y}} p_y \|\eta_y - \overline{\eta}\|_2^2 \leq \sum_{y \in \mathcal{Y}} 2 p_y \left( \|\eta\|_2^2 + \sum_{y' \in \mathcal{Y}} p_{y'} \|\eta_{y'}\|_2^2 \right) \\
&= 4 \sum_{y \in \mathcal{Y}} p_y \|\eta_y\|_2^2 \leq 4\|\eta\|_2^2
\end{aligned}
$$

So we have $\|A\|_{op} \leq 2\|\eta\|_2$. $\qquad\square$

**Lemma 10.** *If parameter $\tilde{\theta} = \{\hat{\lambda}, \eta\}$ satisfies that $\|\eta\|_2 \leq \frac{\delta}{4\beta}$, then $V(D; \tilde{\theta}) \leq \delta$, where $V(D; \theta)$ is defined in Eq. (16).*

*Proof.* Let $S_L = \{\Gamma^{(1)}, \dots, \Gamma^{(L)}\}$, and let $\mathbf{H}^{(l)}$ and $\mathbf{h}^{(l)}$ be short for $\mathbf{H}^{(l)}(X, S_l; \tilde{\theta})$ and $\mathbf{h}^{(l)}(X, \mathrm{E}(S_l); \tilde{\theta})$, respectively.

From lemma 9, we have $\|f_L(x; \eta) - f_L(y; \eta)\|_2 \leq 2\|\eta\|_2 \|x - y\|_2$. Then,

$$
\begin{aligned}
\left\| \mathrm{E}_{S_L} \left[ \mathbf{H}^L \right] - \mathbf{h}^L \right\|_2 &= \left\| \mathrm{E}_{S_{L-1}} \left[ f_L(\mathbf{H}^{(L-1)}; \eta) \right] - f_L(\mathbf{h}^{(L-1)}; \eta) \right\|_2 \\
&\leq \mathrm{E}_{S_{L-1}} \left\| f_L(\mathbf{H}^{(L-1)}; \eta) - f_L(\mathbf{h}^{(L-1)}; \eta) \right\|_2 \\
&\leq 2\|\eta\|_2 \left\| \mathbf{H}^{(L-1)} - \mathbf{h}^{(L-1)} \right\|_2 \\
&\leq 4\beta\|\eta\|_2 \leq \delta
\end{aligned}
$$

$\qquad\square$

Lemma 10 says that we can get $\theta$ satisfying the expectation-linearization constrain by explicitly scaling down $\hat{\eta}$ while keeping $\hat{\lambda}$.

In order to prove Theorem 5, we make the following assumptions:

- The dimension of $\mathbf{h}^{(L-1)}$ is $d$, i.e. $\mathbf{h}^{(L-1)} \in \mathcal{R}^d$.
- Since $\forall y \in \mathcal{Y}, p(y|x; \hat{\theta}) > 0$, we assume $p(y|x; \hat{\theta}) \geq 1/b$, where $b \geq |\mathcal{Y}| = k$.
- As in the body text, let $p(y|x, s; \hat{\theta})$ be nonuniform, and in particular let
$\hat{\eta}_{y^*}^T \mathbf{h}^{(L-1)}(x, s; \hat{\lambda}) - \hat{\eta}_y^T \mathbf{h}^{(L-1)}(x, s; \hat{\lambda}) > c\|\hat{\eta}\|_2, \forall y \neq y^*$.

For convenience, we denote $\eta^T \mathbf{h}^{(L-1)}(x, s; \lambda) = \eta^T u_y(x, s; \lambda)$, where $u_y^T(x, s; \lambda) = (v_0^T, \dots, v_k^T)$ and

$$
v_i = \begin{cases} \mathbf{h}^{(L-1)}(x, s; \lambda) & \text{if } i = y \\ 0 & \text{otherwise} \end{cases}
$$

To prove Theorem 5, we first prove the following lemmas.

**Lemma 11.** *If $p(y|x; \hat{\theta}) \geq 1/b$, then $\forall \alpha \in [0, 1]$, for parameter $\tilde{\theta} = \{\hat{\lambda}, \alpha\hat{\eta}\}$, we have*

$$p(y|x; \tilde{\theta}) \geq \frac{1}{b}$$

*Proof.* We define

$$f(\alpha) \triangleq (y|x,s;\tilde{\theta}) = \frac{e^{\alpha \eta_y^T \mathbf{h}^{(L-1)}(x,s;\hat{\lambda})}}{\sum\limits_{y' \in \mathcal{Y}} e^{\alpha \eta_{y'}^T \mathbf{h}^{(L-1)}(x,s;\hat{\lambda})}} = \frac{\left(e^{\eta_y^T \mathbf{h}^{(L-1)}(x,s;\hat{\lambda})}\right)^\alpha}{\sum\limits_{y' \in \mathcal{Y}} \left(e^{\eta_{y'}^T \mathbf{h}^{(L-1)}(x,s;\hat{\lambda})}\right)^\alpha}$$

Since $\mathcal{Y} = \{1, \ldots, k\}$, for fixed $x \in \mathcal{X}, s \in \mathcal{S}$, $\log f(\alpha)$ is a concave function w.r.t $\alpha$. Since $b \geq k$, we have

$$\log f(\alpha) \geq (1 - \alpha) \log f(0) + \alpha \log f(1) \geq -\log b$$

So we have $\forall x, s, p(y|x,s;\tilde{\theta}) \geq 1/b$. Then

$$p(y|x;\tilde{\theta}) = \mathrm{E}_S \left[ p(y|x,S;\hat{\theta}) \right] \geq \frac{1}{b}$$

$\square$

**Lemma 12.** *if $y$ is not the majority class, i.e. $y \neq y^*$, then for parameter $\tilde{\theta} = \{\hat{\lambda}, \alpha \hat{\eta}\}$*

$$p(y|x,s,\tilde{\theta}) \leq e^{-c\alpha \|\hat{\eta}\|_2}$$

*Proof.*

$$p(y|x,s,\tilde{\theta}) = \frac{e^{\alpha \hat{\eta}^T u_y}}{\sum\limits_{y' \in \mathcal{Y}} e^{\alpha \hat{\eta}^T u_{y'}}} \leq \frac{e^{\alpha \hat{\eta}^T u_y}}{e^{\alpha \hat{\eta}^T u_{y^*}}} \leq e^{-c\alpha \|\hat{\eta}\|_2}$$

$\square$

**Lemma 13.** *For a fixed $x$ and $s$, the absolute value of the entry of the vector under the parameter $\tilde{\theta} = \{\hat{\lambda}, \alpha \hat{\eta}\}$:*

$$|p(y|x,s;\tilde{\theta})(u_y - \mathrm{E}_Y[u_Y])|_i \leq \beta(k-1)e^{-c\alpha \|\hat{\eta}\|_2}$$

*Proof.* Suppose $y$ is the majority class of $p(y|x,s;\tilde{\theta})$. Then,

$$u_y - \mathrm{E}_y[u_Y] = (v_{y'})_{y'=1}^k$$

where

$$v_y = \begin{cases} (1 - p(y|x,s;\tilde{\theta}))\mathbf{h}^{(L-1)} & \text{if } y = y^* \\ -p(y|x,s;\tilde{\theta})\mathbf{h}^{(L-1)} & \text{otherwise} \end{cases}$$

From Lemma 12, we have

$$|p(y|x,s;\tilde{\theta})(u_y - \mathrm{E}_Y[u_Y])|_i \leq |(u_y - \mathrm{E}_Y[u_Y])|_i \leq \beta(k-1)e^{-c\alpha \|\hat{\eta}\|_2}$$

Now, we suppose $y$ is not the majority class of $p(y|x,s;\tilde{\theta})$. Then,

$$|p(y|x,s;\tilde{\theta})(u_y - \mathrm{E}_Y[u_Y])|_i \leq p(y|x,s;\tilde{\theta})\beta \leq \beta e^{-c\alpha \|\hat{\eta}\|_2}$$

Overall, the lemma follows. $\square$

**Lemma 14.** *We denote the matrix*

$$\begin{aligned} A \triangleq \quad & \mathrm{E}_S \left[ \frac{p(y|x,s;\tilde{\theta})}{p(y|x;\tilde{\theta})}(u_y - \mathrm{E}_Y[u_Y])(u_y - \mathrm{E}_Y[u_Y])^T \right] \\ & -\mathrm{E}_S \left[ \frac{p(y|x,s;\tilde{\theta})}{p(y|x;\tilde{\theta})}(u_y - \mathrm{E}_Y[u_Y]) \right] \mathrm{E}_S \left[ \frac{p(y|x,s;\tilde{\theta})}{p(y|x;\tilde{\theta})}(u_y - \mathrm{E}_Y[u_Y]) \right]^T \end{aligned}$$

*Then the absolute value of the entry of $A$ under the parameter $\tilde{\theta} = \{\hat{\lambda}, \alpha \hat{\eta}\}$:*

$$|A_{ij}| \leq 2b(k-1)\beta^2 e^{-c\alpha \|\hat{\eta}\|_2}$$

*Proof.* From Lemma 11, we have $p(y|x; \tilde{\theta}) \geq 1/b$. Additionally, the absolute value of the entry of $u_y - \mathrm{E}_Y[u_Y]$ is bounded by $\beta$. We have for each $i$

$$\left| \mathrm{E}_S \left[ \frac{p(y|x, s; \tilde{\theta})}{p(y|x; \tilde{\theta})} (u_y - \mathrm{E}_Y[u_Y]) \right] \right|_i \leq \mathrm{E}_S \left[ \frac{p(y|x, s; \tilde{\theta})}{p(y|x; \tilde{\theta})} \beta \right] = \beta$$

Then from Lemma 13

$$|A_{ij}| \leq 2b(k-1)\beta^2 e^{-c\alpha\|\hat{\eta}\|_2}$$

$\square$

**Lemma 15.** *We denote the matrix*

$$B \triangleq \mathrm{E}_S \left[ \frac{p(y|x, s; \tilde{\theta})}{p(y|x; \tilde{\theta})} \left( \mathrm{E}_Y \left[ u_Y u_Y^T \right] - \mathrm{E}_Y[u_Y] \mathrm{E}_Y[u_Y]^T \right) \right]$$

*Then the absolute value of the entry of $B$ under the parameter $\tilde{\theta} = \{\hat{\lambda}, \alpha\hat{\eta}\}$:*

$$|B_{ij}| \leq 2(k-1)\beta^2 e^{-c\alpha\|\hat{\eta}\|_2}$$

*Proof.* We only need to prove that for fixed $x$ and $s$, for each $i, j$:

$$\left| \mathrm{E}_Y \left[ u_Y u_Y^T \right] - \mathrm{E}_Y[u_Y] \mathrm{E}_Y[u_Y]^T \right|_{ij} \leq 2(k-1)\beta^2 e^{-c\alpha\|\hat{\eta}\|_2}$$

Since

$$\left| \mathrm{E}_Y \left[ u_Y u_Y^T \right] - \mathrm{E}_Y[u_Y] \mathrm{E}_Y[u_Y]^T \right|_{ij} = |\mathrm{Cov}_Y[(u_Y)_i, (u_Y)_j]| \leq \beta^2 \sum_{y=1}^k p(y|x, s; \tilde{\theta}) - p(y|x, s; \tilde{\theta})^2$$

Suppose $y$ is the majority class. Then from Lemma 12,

$$p(y|x, s; \tilde{\theta}) - p(y|x, s; \tilde{\theta})^2 \leq 1 - p(y|x, s; \tilde{\theta}) \leq (k-1)e^{-c\alpha\|\hat{\eta}\|_2}$$

If $y$ is not the majority class. Then,

$$p(y|x, s; \tilde{\theta}) - p(y|x, s; \tilde{\theta})^2 \leq p(y|x, s; \tilde{\theta}) \leq e^{-c\alpha\|\hat{\eta}\|_2}$$

So we have

$$\sum_{y=1}^k p(y|x, s; \tilde{\theta}) - p(y|x, s; \tilde{\theta})^2 \leq 2(k-1)e^{-c\alpha\|\hat{\eta}\|_2}$$

The lemma follows. $\square$

**Lemma 16.** *Under the parameter $\tilde{\theta} = \{\hat{\lambda}, \alpha\hat{\eta}\}$, the largest eigenvalue of the matrix*

$$\frac{1}{n} \sum_{i=1}^n \left( A(x_i, y_i) - B(x_i, y_i) \right) \tag{8}$$

*is at most*

$$2dk(k-1)(b+1)\beta^2 e^{-c\alpha\|\hat{\eta}\|_2}$$

*Proof.* From Lemma 14 and Lemma 15, each entry of the matrix in (8) is at most $2(k-1)(b+1)\beta^2 e^{-c\alpha\|\hat{\eta}\|_2}$. Thus, by Gershgorin's theorem, the maximum eigenvalue of the matrix in (8) is at most $2dk(k-1)(b+1)\beta^2 e^{-c\alpha\|\hat{\eta}\|_2}$. $\square$

Now, we can prove Theorem 5 by constructing a scaled version of $\hat{\theta}$ that satisfies the expectation-linearization constraint.

**Proof of Theorem 5**

*Proof.* Consider the likelihood evaluated at $\tilde{\theta} = \{\hat{\lambda}, \alpha\hat{\eta}\}$, where $\alpha = \frac{\delta}{4\beta\|\hat{\eta}\|_2}$. If $\alpha > 1$, then $\|\eta\|_2 > \frac{\delta}{4\beta}$. We know the MLE $\hat{\theta}$ already satisfies the expectation-linearization constraint. So we can assume that $0 \leq \alpha \leq 1$, and we know that $\tilde{\theta}$ satisfies $V(D; \tilde{\theta}) \leq \delta$. Then,

$$\Delta_l(\hat{\theta}, \hat{\theta}_\delta) \leq \Delta_l(\hat{\theta}, \tilde{\theta}) = \frac{1}{n}(l(D; \hat{\theta}) - l(D; \tilde{\theta})) = g(\hat{\lambda}, \hat{\eta}) - g(\hat{\lambda}, \alpha\hat{\eta})$$

where $g(\lambda, \eta) = \frac{1}{n}l(D; (\lambda, \eta))$. Taking the second-order Taylor expansion about $\eta$, we have

$$g(\hat{\lambda}, \alpha\hat{\eta}) = g(\hat{\lambda}, \hat{\eta}) + \nabla_\eta^T g(\hat{\lambda}, \hat{\eta})(\alpha\hat{\eta} - \hat{\eta}) + (\alpha\hat{\eta} - \hat{\eta})^T \nabla_\eta^2 g(\hat{\lambda}, \hat{\eta})(\alpha\hat{\eta} - \hat{\eta})$$

Since $\hat{\theta}$ is the MLE, the first-order term $\nabla_\eta^T g(\hat{\lambda}, \hat{\eta})(\alpha\hat{\eta} - \hat{\eta}) = 0$. The Hessian in the second-order term is just Eq.(8). Thus, from Lemma 16 we have

$$
\begin{aligned}
g(\hat{\lambda}, \alpha\hat{\eta}) &\leq g(\hat{\lambda}, \hat{\eta}) - (1 - \alpha)^2 \|\hat{\eta}\|_2^2 2dk(k-1)(b+1)\beta^2 e^{-c\alpha\|\hat{\eta}\|_2} \\
&= g(\hat{\lambda}, \hat{\eta}) - 2dk(k-1)(b+1)\beta^2 \left(\|\hat{\eta}\|_2 - \frac{\delta}{4\beta}\right)^2 e^{-c\delta/4\beta} \\
&= g(\hat{\lambda}, \hat{\eta}) - c_1\beta^2 \left(\|\hat{\eta}\|_2 - \frac{\delta}{4\beta}\right)^2 e^{-c_2\delta/4\beta}
\end{aligned}
$$

with setting $c1 = 2dk(k-1)(b+1)$ and $c2 = c$. Then the theorem follows. $\square$

## C.3 PROOF OF THEOREM 6: UNIFORM BOUND OF MODEL ACCURACY

In the following, we denote $\tilde{\theta} = \{\hat{\lambda}, \alpha\hat{\eta}\}$.

**Lemma 17.** *For each $y \in \mathcal{Y}$, if $p(y|x, s; \hat{\theta}) \geq 1/k$, then $\forall \alpha \in [0, 1]$*

$$p(y|x, s; \tilde{\theta}) \geq \frac{1}{k}$$

*Proof.* This lemma can be regarded as a corollary of Lemma 11. $\square$

**Lemma 18.** *For a fixed $x$ and $s$, we denote $e^{\hat{\eta}_y^T \mathbf{h}^{(L-1)}(x, s; \hat{\lambda})} = w_y$. Then we have*

$$p(y|x, s; \tilde{\theta}) = \frac{e^{\alpha\hat{\eta}_y^T \mathbf{h}^{(L-1)}(x, s; \hat{\lambda})}}{\sum\limits_{y' \in \mathcal{Y}} e^{\alpha\hat{\eta}_{y'}^T \mathbf{h}^{(L-1)}(x, s; \hat{\lambda})}} = \frac{(w_y)^\alpha}{\sum\limits_{y' \in \mathcal{Y}} (w_{y'})^\alpha}$$

*Additionally, we denote $g_s(\alpha) = \sum\limits_{y' \in \mathcal{Y}} p(y'|x, s; \tilde{\theta}) \log w_{y'} - \log w_y$. We assume $g_s(0) \geq 0$. Then we have $\forall \alpha \geq 0$*

$$g_s(\alpha) \geq 0$$

*Proof.*

$$\frac{\partial g_s(\alpha)}{\partial \alpha} = \sum_{y' \in \mathcal{Y}} \log w_{y'} \frac{\partial p(y'|x, s; \tilde{\theta})}{\partial \alpha} = \mathrm{Var}_Y\left[\log w_Y | X - x, S = s\right] \geq 0$$

So $g_s(\alpha)$ is non-decreasing. Since $g_s(0) \geq 0$, we have $g_s(\alpha) \geq 0$ when $\alpha \geq 0$. $\square$

From above lemma, we have for each training instance $(x_i, y_i) \in D$, and $\forall \alpha \in [0, 1]$,

$$\mathrm{E}_Y\left[\log p(Y|x_i, s; \tilde{\theta})\right] \geq \log p(y_i|x_i, s; \tilde{\theta}) \tag{9}$$

For convenience, we define

$$m(s, y) = \log p(y|x, s; \tilde{\theta}) - \mathrm{E}_Y\left[\log p(Y|x, s; \tilde{\theta})\right]$$

**Lemma 19.** *If $y$ satisfies Lemma 17 and $g_s(\alpha) \geq 0$, then*

$$\mathrm{Var}_Y[m(s, Y)] \geq m(s, y)^2$$

*Proof.* First we have

$$m(s, y) = \log p(y|x, s; \tilde{\theta}) - \log 1/k - KL\left(p(\cdot|x, s; \tilde{\theta})|\mathrm{Unif}(\mathcal{Y})\right) \leq 0$$

So we have

$$
\begin{aligned}
(\mathrm{Var}_Y[m(s, Y)])^{1/2} &= \sqrt{\mathrm{E}_Y\left[\left(\log p(Y|x, s; \tilde{\theta}) - \mathrm{E}_Y\left[\log p(Y|x, s; \tilde{\theta})\right]\right)^2\right]} \\
&\geq \mathrm{E}_Y\left[\left|\log p(Y|x, s; \tilde{\theta}) - \mathrm{E}_Y\left[\log p(Y|x, s; \tilde{\theta})\right]\right|\right] \\
&= \mathrm{E}_Y\left[\left|KL\left(p(\cdot|x, s; \tilde{\theta})|\mathrm{Unif}(\mathcal{Y})\right) + \log 1/k - \log p(Y|x, s; \tilde{\theta})\right|\right] \\
&= \mathrm{E}_Y\left[KL\left(p(\cdot|x, s; \tilde{\theta})|\mathrm{Unif}(\mathcal{Y})\right) + \left|\log 1/k - \log p(Y|x, s; \tilde{\theta})\right|\right] \\
&\geq KL\left(p(\cdot|x, s; \tilde{\theta})|\mathrm{Unif}(\mathcal{Y})\right) + \mathrm{E}_Y\left[\log p(Y|x, s; \tilde{\theta}) - \log 1/k\right] \\
&= 2KL\left(p(\cdot|x, s; \tilde{\theta})|\mathrm{Unif}(\mathcal{Y})\right)
\end{aligned}
$$

As $KL\left(p(\cdot|x, s; \tilde{\theta})|\mathrm{Unif}(\mathcal{Y})\right) \geq 0$ and $\log p(y|x, s; \tilde{\theta}) \geq \log 1/k$. So we have

$$2KL\left(p(\cdot|x, s; \tilde{\theta})|\mathrm{Unif}(\mathcal{Y})\right) \geq KL\left(p(\cdot|x, s; \tilde{\theta})|\mathrm{Unif}(\mathcal{Y})\right) + \log 1/k - \log p(y|x, s; \tilde{\theta}) = -m(s, y)$$

Then the lemma follows. $\square$

From Lemma 19 and Eq. (9), we have for each training instance $(x_i, y_i) \in D$, and $\forall \alpha \in [0, 1]$,

$$\mathrm{Var}_Y[m(s, Y)] \geq m(s, y_i)^2 \tag{10}$$

**Lemma 20.** *For each training instance $(x_i, y_i) \in D$, and $\forall \alpha \in [0, 1]$, we have*

$$\log p(y_i|x_i; \{\hat{\lambda}, \alpha\hat{\eta}\}) \geq (1 - \alpha) \log p(y_i|x_i; \{\hat{\lambda}, 0\}) + \alpha \log p(y_i|x_i; \{\hat{\lambda}, \hat{\eta}\})$$

*Proof.* We define

$$f(\alpha) = \log p(y_i|x_i; \{\hat{\lambda}, \alpha\hat{\eta}\}) - (1 - \alpha) \log p(y_i|x_i; \{\hat{\lambda}, 0\}) - \alpha \log p(y_i|x_i; \{\hat{\lambda}, \hat{\eta}\})$$

Because $f(0) = f(1) = 0$, we only need to prove that $f(\alpha)$ is concave on $[0, 1]$. We have

$$\nabla^2 f(\alpha) = -\mathrm{E}_{S|Y=y_i}[\mathrm{Var}_Y[m(S, Y)]] + \mathrm{Var}_{S|Y=y_i}[m(S, y_i)]$$

where $S|Y = y_i$ is under the probability distribution $p(s|Y = y_i, x_i; \tilde{\theta}) = \frac{p(y_i|x_i, S; \tilde{\theta})p(s)}{p(y_i|x_i; \tilde{\theta})}$
From Eq. (10), we have

$$\mathrm{E}_{S|Y=y_i}[\mathrm{Var}_Y[m(S, Y)]] \geq \mathrm{E}_{S|Y=y_i}[m(S, y_i)^2] \geq \mathrm{Var}_{S|Y=y_i}[m(S, y_i)]$$

So we have $\nabla^2 f(\alpha) \leq 0$. The lemma follows. $\square$

Now, we can prove Theorem 6 by using the same construction of an expectation-linearizing parameter as in Theorem 5.

**Proof of Theorem 6**

*Proof.* Consider the same parameter $\tilde{\theta} = \{\hat{\lambda}, \alpha\hat{\eta}\}$, where $\alpha = \frac{\delta}{4\beta\|\hat{\eta}\|_2} \leq 1$. we know that $\tilde{\theta}$ satisfies $V(D; \tilde{\theta}) \leq \delta$. Then,

$$\Delta_l(\hat{\theta}, \hat{\theta}_\delta) \leq \Delta_l(\hat{\theta}, \tilde{\theta}) = \frac{1}{n}(l(D; \hat{\theta}) - l(D; \tilde{\theta}))$$

From Lemma 20 we have:

$$l(D; \tilde{\theta}) = l(D; \{\hat{\lambda}, \alpha\hat{\eta}\}) \geq (1 - \alpha)l(D; \{\hat{\lambda}, 0\}) + \alpha l(D; \{\hat{\lambda}, \hat{\eta}\})$$

So

$$
\begin{aligned}
\Delta_l(\hat{\theta}, \hat{\theta}_\delta) &\leq (1 - \alpha)\frac{1}{n}\left(l(D; \hat{\theta}) - l(D; \{\hat{\lambda}, 0\})\right) \\
&= (1 - \alpha)\frac{1}{n}\sum_{i=1}^{n} \log p(y_i|x_i; \hat{\theta}) - \log \mathrm{Unif}(\mathcal{Y}) \\
&\asymp (1 - \alpha)\mathrm{E}\left[\mathrm{KL}\left(p(\cdot|X; \theta)\|\mathrm{Unif}(\mathcal{Y})\right)\right] \\
&\leq \left(1 - \frac{\delta}{4\beta\|\hat{\eta}\|_2}\right)\mathrm{E}\left[\mathrm{KL}\left(p(\cdot|X; \theta)\|\mathrm{Unif}(\mathcal{Y})\right)\right]
\end{aligned}
$$

$\square$

# D    DETAILED DESCRIPTION OF EXPERIMENTS

## D.1    NEURAL NETWORK ARCHITECTURES

**MNIST**    For MNIST, we train 6 different fully-connected (dense) neural networks with 2 or 3 layers (see Table 1). For all architectures, we used dropout rate $p = 0.5$ for all hidden layers and $p = 0.2$ for the input layer.

**CIFAR-10 and CIFAR-100**    For the two CIFAR datasets, we used the same architecture in Srivastava et al. (2014) — three convolutional layers followed by two fully-connected hidden layers. The convolutional layers have 96, 128, 265 filters respectively, with a $5 \times 5$ receptive field applied with a stride of 1. Each convolutional layer is followed by a max pooling layer pools $3 \times 3$ regions at strides of 2. The fully-connected layers have 2048 units each. All units use the rectified linear activation function. Dropout was applied to all the layers with dropout rate $p = (0.1, 0.25, 0.25, 0.5, 0.5, 0.5)$ for the layers going from input to convolutional layers to fully-connected layers.

## D.2    NEURAL NETWORK TRAINING

Neural network training in all the experiments is performed with mini-batch stochastic gradient descent (SGD) with momentum. We choose an initial learning rate of $\eta_0$, and the learning rate is updated on each epoch of training as $\eta_t = \eta_0/(1 + \rho t)$, where $\rho$ is the decay rate and $t$ is the number of epoch completed. We run each experiment with 2,000 epochs and choose the parameters achieving the best performance on validation sets.

Table 3 summarizes the chosen hyper-parameters for all experiments. Most of the hyper-parameters are chosen from Srivastava et al. (2014). But for some experiments, we cannot reproduce the performance reported in Srivastava et al. (2014) (We guess one of the possible reasons is that we used different library for implementation.). For these experiments, we tune the hyper-parameters on the validation sets by random search. Due to time constrains it is infeasible to do a random search across the full hyper-parameter space. Thus, we try to use as many hyper-parameters reported in Srivastava et al. (2014) as possible.

## D.3    EFFECT OF EXPECTATION-LINEARIZATION RATE $\lambda$

Table 4 illustrates the detailed results of the experiments on the effect of $\lambda$. For MNIST, it lists the error rates under different $\lambda$ values for six different network architectures. For two datasets of CIFAR, it gives the error rates under different $\lambda$ values, among with the empirical expectation-linearization risk $\hat{\Delta}$.

Table 3: Hyper-parameters for all experiments.

| Experiment | Hyper-parameter | | |
|---|---|---|---|
| MNIST | batch size | 200 | |
| | initial learning rate $\eta_0$ | 0.1 | |
| | decay rate $\rho$ | 0.025 | |
| | momentum | 0.9 | |
| | momentum type | standard | |
| | max-norm constrain | 3.5 | |
| CIFAR | | **10** | **100** |
| | batch size | 100 | 100 |
| | initial learning rate $\eta_0$ for conv layers | 0.001 | 0.001 |
| | initial learning rate $\eta_0$ for dense layers | 0.1 | 0.02 |
| | decay rate $\rho$ | 0.005 | 0.005 |
| | momentum | 0.95 | 0.95 |
| | momentum type | standard | nesterov |
| | max-norm constrain | 4.0 | 2.0 |
| | L2-norm decay | 0.001 | 0.001 |

Table 4: Detailed results for experiments on the effect of $\lambda$.

| Experiment | | $\lambda$ | | | | | | | |
|---|---|---|---|---|---|---|---|---|---|
| | | 0.0 | 0.5 | 1.0 | 2.0 | 3.0 | 5.0 | 7.0 | 10.0 |
| MNIST | model 1 | 1.23 | 1.12 | 1.12 | 1.08 | 1.07 | 1.10 | 1.25 | 1.35 |
| | model 2 | 1.19 | 1.14 | 1.08 | 1.04 | 1.03 | 1.07 | 1.13 | 1.21 |
| | model 3 | 1.05 | 1.04 | 0.98 | 1.03 | 1.05 | 1.05 | 1.10 | 1.12 |
| | model 4 | 1.07 | 1.02 | 0.97 | 0.94 | 0.96 | 1.01 | 1.05 | 1.20 |
| | model 5 | 1.03 | 0.95 | 0.95 | 0.90 | 0.92 | 0.98 | 1.03 | 1.08 |
| | model 6 | 0.99 | 0.98 | 0.93 | 0.87 | 0.96 | 0.98 | 1.05 | 1.10 |
| | | $\lambda$ | | | | | | | |
| | | 0.0 | 0.1 | 0.5 | 1.0 | 2.0 | 5.0 | 10.0 | |
| CIFAR-10 | error rate | 12.82 | 12.52 | 12.38 | 12.20 | 12.60 | 12.84 | 13.10 | |
| | $\hat{\Delta}$ | 0.0139 | 0.0128 | 0.0104 | 0.0095 | 0.0089 | 0.0085 | 0.0077 | |
| CIFAR-100 | error rate | 37.22 | 36.75 | 36.25 | 37.01 | 37.18 | 37.58 | 38.01 | |
| | $\hat{\Delta}$ | 0.0881 | 0.0711 | 0.0590 | 0.0529 | 0.0500 | 0.0467 | 0.0411 | |

