# Peer review of "Dropout with Expectation-linear Regularization"

_ICLR 2017 — accepted_

[Official Review · AnonReviewer3 · rating 8 · confidence 3 · 16 Dec 2016]
**Good paper**

This paper puts forward a not entirely new, but also not sufficiently understood interpretation of dropout regularization. The authors derive useful theorems that estimate or put bounds on key quantities that are of interest when analyzing dropout regularized networks from their perspective. They furthermore introduce an explicit regularization term that should have a well understood impact on these key quantities. In the experimental section they convincingly show that the proposed regularization indeed has the expected effect and that their perspective on dropout is therefore useful and meaningful.

Their proposed regularization also seems to have a positive impact on the models performance but they demonstrate this only on rel. small scale benchmark problems. I therefore don’t belief that this approach will have a large impact on how practitioner train models.  But their general perspective is well aligned with the recently proposed idea of “Dropout as a bayesian approximation” and the insights and theorems in this paper might enable future work in that direction.

[Official Review · AnonReviewer2 · rating 7 · confidence 4 · 17 Dec 2016]
**No Title**

This paper introduces dropout as a latent variable model (LVM). Leveraging this formulation authors analyze the dropout “inference gap” which they define to be the gap between network output during training (where an instance of dropout is used for every training sample) and test (where expected dropout values are used to scale node outputs).  They introduce the notion of expectation linearity and use this to derive bounds on the inference gap under some (mild) assumptions.  Furthermore, they propose use of per-sample based inference gap as a regularizer, and present analysis of accuracy of models with expectation-linearization constraints as compared to those without.

One relatively minor issue I see with the LVM view of dropout is that it seems applicable only to probabilistic models whereas dropout is more generally applicable to deep networks.  However I’d expect that the regularizer formulation of dropout would be effective even in non-probabilistic models.

MC dropout on page 8 is not defined, please define.

On page 9 it is mentioned that with the proposed regularizer the standard dropout networks achieve better results than when Monte Carlo dropout is used.  This seems to be the case only on MNIST dataset and not on CIFAR?

From Tables 1 and 2 it also appears that MC dropout achieves best performance across tasks and methods but it is of course an expensive procedure.  Comments on the computational efficiency of various dropout procedures - to go with the accuracy results - would be quite valuable.

Couple of typos:
- Pg. 2 “ … x is he input …” -> “ … x is the input …”
- Pg. 5 “ … as defined in (1), is …” -> ref. to (1) is not right at two places in this paragraph

Overall it is a good paper, I think should be accepted and discussed at the conference.

[Official Review · AnonReviewer1 · rating 8 · confidence 3 · 23 Dec 2016]
**Good paper**

summary

The paper explains dropout with a latent variable model where the dropout variable (0 or 1 depending on which units should be dropped) is not observed and is accordingly marginalised. Maximum likelihood under this model is not tractable but standard dropout then corresponds to a simple Monte Carlo approximation of ML for this model.

The paper then introduces a theoretical framework for analysing the discrepancy (called inference gap) between the model at training (model ensemble, or here the latent variable model), and the model at testing (where usually what should be an expectation over the activations over many models becomes the activation of one model with averaged weights).
This framework introduces several notions (e.g. expectation linearity) which allow the study of which transition functions (and more generally layers) can have a small inference gap. Theorem 3 gives a bound on the inference gap.

Finally a new regularisation term is introduced to account for minimisation of the inference gap during learning.

Experiments are performed on MNIST, CIFAR-10 and CIFAR-100 and show that the method has the potential to perform better than standard dropout and at the level of Monte Carlo Dropout (the standard method to compute the real dropout outputs consistently with the training assumption of an ensemble, of course quite expensive computationally)


The study gives a very interesting theoretical model for dropout as a latent variable model where standard dropout is then a monte carlo approximation. This is very probably widely applicable to further studies of dropout.

The framework for the study of the inference gap is interesting although maybe somewhat less widely applicable.

The proposed model is convincing although 1. it is tested on simple datasets 2. the gains are relatively small and 3. there is an increased computational cost during training because a new hyper-parameter is introduced.

p6 line 8 typo: expecatation

[Author Response · Xuezhe Ma · 01 Jan 2017]
**Revision of the paper**

We made the following revisions:

1. We switched the section 6.3 and 6.4 to make the paper more clear.

2. We added the definition of MC dropout on page 8.

3. We fixed all the typos in the three reviewers' comments.

[Final Decision · Program Chairs · 06 Feb 2017]
**ICLR committee final decision**

This paper presents a theoretical underpinning of dropout, and uses this derivation to both characterize its properties and to extend the method. A solid contribution. I am surprised that none of the reviewers mentioned that this work is closely related to the uncited 2015 paper "Variational Dropout and the Local Reparameterization Trick" by Diederik P. Kingma, Tim Salimans, Max Welling.